# Herbicide Resistance to Metsulfuron-Methyl in *Rumex dentatus* L. in North-West India and Its Management Perspectives for Sustainable Wheat Production

Ankur Chaudhary [1,*], Rajender Singh Chhokar [2], Sachin Dhanda [3], Prashant Kaushik [4], Simerjeet Kaur [5], Todar Mal Poonia [3], Rajbir Singh Khedwal [3], Surender Kumar [3] and Satbir Singh Punia [3]

1   Regional Research Station, Chaudhary Charan Singh Haryana Agricultural University, Uchani 132001, Haryana, India
2   ICAR-Indian Institute of Wheat and Barley Research, Karnal 132001, Haryana, India; rs_chhokar@yahoo.co.in
3   Department of Agronomy, Chaudhary Charan Singh Haryana Agricultural University, Hisar 125004, Haryana, India; sachindhanda016@gmail.com (S.D.); todarmal.poonia6@gmail.com (T.M.P.); rajbirsinghkhedwal1524@gmail.com (R.S.K.); surenderdhanday1984@gmail.com (S.K.); puniasatbir@gmail.com (S.S.P.)
4   Instituto de Conservación y Mejora de la Agrodiversidad Valenciana, Universitat Politècnica de València, 46022 Valencia, Spain; prakau@doctor.upv.es
5   Department of Agronomy, Punjab Agricultural University, Ludhiana 141004, Punjab, India; simer@pau.edu
*   Correspondence: ankurchaudhary@hau.ac.in

**Abstract:** Herbicide resistance in weeds is a global threat to sustaining food security. In India, herbicide-resistant *Phalaris minor* was the major problem in wheat for more than two decades, but the continuous use of metsulfuron-methyl (an ALS inhibitor) to control broadleaf weeds has resulted in the evolution of ALS inhibitor-resistant *Rumex dentatus* L. This review summarizes the current scenario of herbicide resistance in *R. dentatus*, along with its ecology and management perspectives. Studies have provided valuable insights on the emergence pattern of *R. dentatus* under different environments in relation to tillage, cropping systems, nutrients, and irrigation. Moreover, *R. dentatus* has exhibited higher emergence under zero tillage, with high infestation levels in rice-wheat compared to other wheat-based cropping systems (sorghum-wheat). Alternative herbicides for the management of resistant *R. dentatus* include pendimethalin, 2,4-D, carfentrazone, isoproturon, and metribuzin. Although the pre-emergence application of pendimethalin is highly successful in suppressing *R. dentatus*, but its efficiency is questionable under lower field soil moisture and heavy residue load conditions. Nevertheless, the biological data may be utilized to control *R. dentatus*. Therefore, herbicide rotation with suitable spray techniques, collecting weed seeds at differential heights from wheat, crop rotation, alternate tillage practices, and straw retention are recommended for addressing the resistance issue in *R. dentatus* in North India conditions. Overall, we discuss the current state of herbicide resistance in *R. dentatus*, the agronomic factors affecting its population, its proliferation in specific cropping systems (rice-wheat), and management strategies for containing an infestation of a resistant population.

**Keywords:** ecology; herbicide resistance; metsulfuron-methyl; *Rumex dentatus*; tillage

## 1. Introduction

Rice-wheat is the most dominant cropping system in India, occupying about 10.3 million ha of the Indo-Gangetic Plains (IGP). The system accounts for 26% of total cereal production and provides 60% total calorie intake [1,2]. The system accounts for approximately 23% and 40% of the total rice and wheat areas, respectively, and accounts for about 85% of total cereal production [3]. Moreover, the rice-wheat cropping system (RWCS) is well proliferated among farmers, with the introduction of fertilizer- and irrigation-responsive short-statured high yielding varieties, development of irrigation facilities, crop-tailored mechanization for

seeding and harvesting, along with encouraging government policies like the minimum support price (MSP) [3–5]. In India, a growth analysis of RWCS demonstrated that area and productivity increased at growth rates of 0.5 and 1.9% for rice and 1.6 and 2.3% for wheat, respectively, from 1950-51 to 2015-16 [6].

Nevertheless, the sustainability of the RWCS and national food security is being halted by the emergence of second-generation problems such as the depletion of good quality groundwater [7], the appearance of multi-nutrient deficiency, extensive residue/straw burning [8,9], the emergence of new insects and pests, diseases, declining factor productivity, and an escalating cost of cultivation [4,10]. Furthermore, the adoption of RWCS also led to a shift in weed flora, with the dominance of some of problematic weeds like *Phalaris minor* and *Rumex dentatus*. In this direction, the accelerated development of herbicide resistance in wheat-associated weeds also limits potential wheat productivity [11].

Earlier, herbicide resistance was reported in *P. minor* in north-western IGP against isoproturon, a substituted urea photosystem II (PS II)-inhibiting herbicide [12–14]. But recently, new weed species among grassy (*Avena ludoviciana* Durieu., *Polypogon monspeliensis* L. Desf.) and broad-leaved weeds (*Chenopodium album* L. and *Rumex dentatus* L.) have evolved resistance against acetolactate synthase (ALS) inhibitor herbicides [11]. Rice and wheat are the two most essential crops from an herbicide industry perspective, as these crops account for about 20 and 28%, respectively, of the total herbicide consumption in India [15].

The short market life of recommended herbicides is a matter of concern as the development of these novel resources requires huge investment. Furthermore, the crisis is aggravated by the fact that no new herbicidal chemistry has been developed or discovered [16]. In this regard, concerted efforts are required towards ecologically based integrated weed management to reduce the resistant biotype selection pressure associated with the single herbicide mode. This review focuses on the current status of herbicide resistance in *Rumex dentatus*, agronomic factors affecting its population, its proliferation in a particular cropping system, and management strategies to modulate the infestation of a resistant population.

## 2. Economic Importance, Biology and Ecology of *Rumex dentatus*

Weeds are the major constraint in the sustainability of crop production, especially of wheat, in IGPs [11,17], as they vigorously compete with crop plants for applied inputs and available resources such as moisture, nutrients, light, and space [17,18]. Weed infestations cause significant yield losses, increases the cost of cultivation, impair produce quality and interfere with farm operations. Weeds also act as a habitat for insects, pests, and diseases. In India, weed infestations in wheat cause a significant yield reduction in productivity to the tune of 20–32% [19]. However, losses to wheat grain yield by weeds depend on the type of weed flora, time of emergence, intensity of weed infestation, level of management practices such as the nature of the wheat cultivar, planting density, sowing method, herbicide application and the level of herbicide resistance [13,17,20]. The monocropping of the RWCS leads to narrow weed profiling of the system, with greater infestation of grass weeds like *Phalaris minor* Retz., *Poa annua* L., Rumex *dentatus* L., and *Medicago denticulata* Willd. as broad-leaved weeds (BLW) [21,22]. The weed flora diversity in wheat varies under different agro-climatic conditions and changes with the nature of the tillage practices, cropping system, and crop management practices being followed [17,21].

Among BLW, toothed dock (*Rumex dentatus*) is a very competitive weed and has the capacity to smother wheat crops [23]. *R. dentatus* is known to severely reduce wheat grain yield by 60% and total dry matter by up to 74% [24]. An increase in *R. dentatus* density from 5 to 30 plants per $m^{-2}$ could reduce wheat grain yield by 2–70% [25,26]. *R. dentatus* competition is reduced the tillers (by 33–41%), shoot biomass (14–36%) and root biomass (37–87%) in wheat varieties (Inqalab 91 and Punjab 96) at 120 days after sowing [27]. A replacement series-based experiment demonstrated that *R. dentatus* had a higher leaf area, specific leaf area, photosynthetic rate, stomatal conductance, and transpiration rate as

compared to wheat and *P. minor*. Subsequently, greater competitiveness was observed in *R. dentatus* against wheat than *P. minor*. In other words, wheat is more susceptible to *R. dentatus* competition than *P. minor* due to the higher relative growth rate and aggressivity index of *R. dentatus* [28]. Additionally, the relative yield total (RYT) of the *R. dentatus* and wheat mixture was more significant than the *P. minor* and wheat mixture. The lower RYT in the latter could be associated with similarity in growth habit (grasses), rooting architecture, and morphology, and their absence in the conditions of the former (*R. dentatus* and wheat mixture) [28].

Globally, the genus Rumex has 200 species, and it is most prevalent in India [29]. *R. dentatus* is a $C_3$ dicot weed of the Polygonaceae family, and it is an annual herb. Its plants at maturity are generally 30–50 cm taller than wheat, with a plant height of about 160 cm with many primary and secondary branches ascending to be almost divaricated (Figure 1). It produces numerous small flowers, is bisexual in nature, and possesses a single whorl of perianth in its panicle racemes [30]. The fruits (16,000 plant$^{-1}$) are single-seeded nuts enclosed within the inner perianth that become enlarged, are winged with toothed margins, and the number of seeds per plant remains same as number of fruits. The inflorescence is racemose; several racemes aggregate to show a panicle-like appearance and contain an abundant starch reserve that constitutes about 21% of the fresh weight of the seed [31].

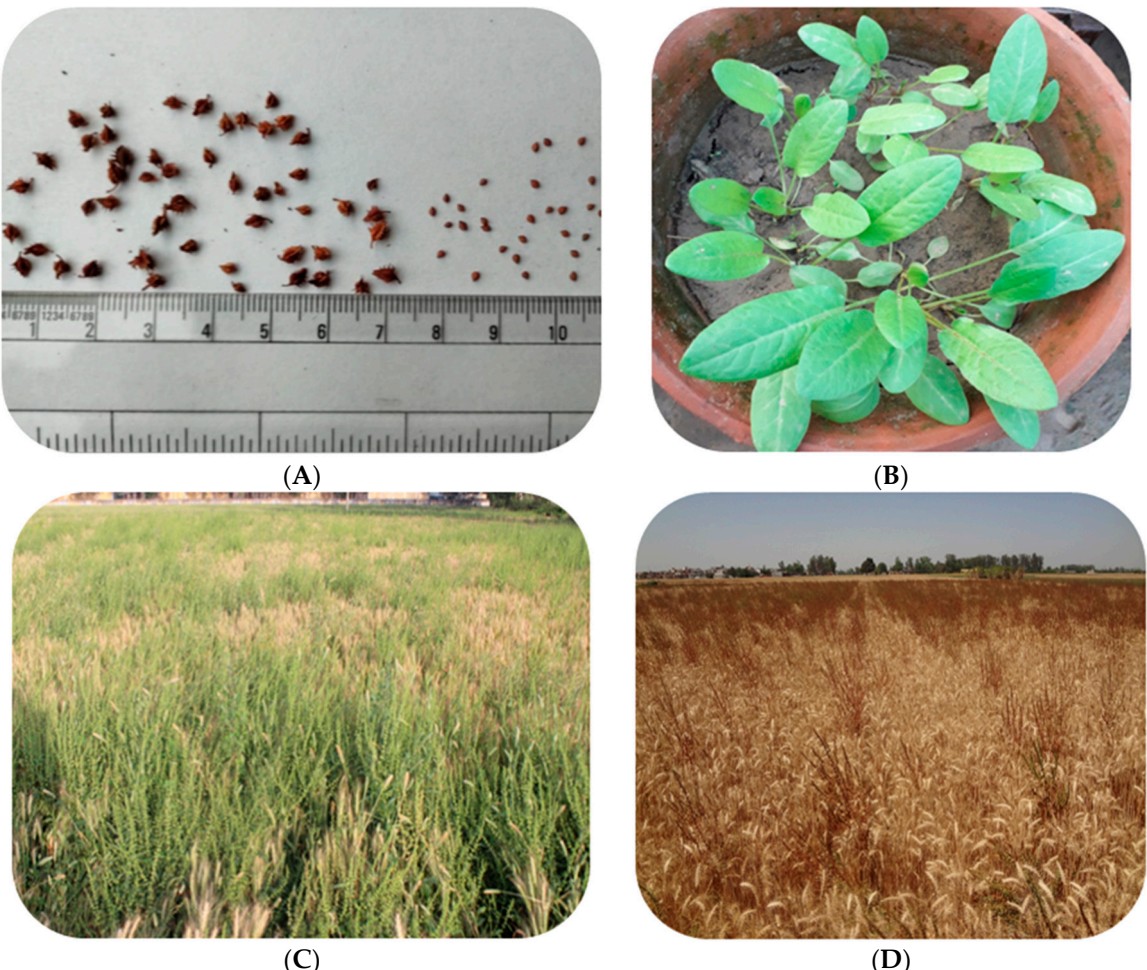

(A)

(B)

(C)

(D)

**Figure 1.** *R. dentatus* seeds (**A**), plants in pot (**B**), infestation in wheat during near maturity (**C**) and maturity (**D**).

Seed germination is a composite physiological process and a combined product of different environmental factors, including seeding, light, soil moisture, temperature, and oxygen [32]. The optimum temperature for germination is around 15 °C [33], however, light failed to influence germination and the seeds showed no dormancy in *R. dentatus*.

Significant germination was recorded, with a temperature of 18/8 °C. The unstratified seeds of *R. dentatus* demonstrated very poor germination at room temperature and under dark conditions. Furthermore, the germination percentage was enhanced by cold stratification and 50 ppm gibberellic acid treatment [34].

The presence of perianth around the seed acts as a barrier to germination. *Rumex* seeds (without perianth) sown at 2 cm depth germinate 35% in one month, but this rate declines to 8% at 4 cm seeding depth [33]. The seed production potential may be as high as 320,000 m$^{-2}$ from a plant population of 8–20 plants m$^{-2}$ [33]. Emergence was similar from the 0 to 1 cm depth, ranging from 61–70%, but was significantly reduced at the 2 cm depth (33%) and only 3% of seeds emerged beyond 4 cm and thereafter (8–16 cm) no emergence was observed [23]. This could be the reason for its greater propensity under zero-tillage (ZT) conditions, as most of the seed remains on the surface with more potential of germination than when buried in conventional tillage.

This surface-based emergence tendency can be used to reduce infestation through tillage manipulation. Following the inversion, tillage will reduce emergence, as seeds lying below 4 cm have an inferior emergence pattern [23,33]. The seeds lying in deeper depths (4 cm), besides having poor germination, will also have poor growth due to their delayed emergence. It has been observed that, at 115 DAS, the plant height of *R. dentatus* at seeding depth 1–2 cm was about 384–558% higher than at 4 cm seeding depth (19 cm). The lower seed weight of *R. dentatus* (1000 seed weight 2.56 g) also suggests it is more of a surface weed the *R. spinosus* (1000 seed weight 12.24 g), which can germinate even beyond a seeding depth of 4 cm [23]. Therefore, tillage practices should be adopted so that the maximum reduction of seed bank can occur either through predation or germination.

## 3. Herbicide Resistance in *Rumex dentatus*

The evolution of resistance in weed accessions against herbicide is a global issue [35] and at present, there are 515 herbicide-resistant weeds (unique cases) in 263 weed species (152 dicots and 111 monocots) [36]. Herbicide resistance in weed accessions has been reported in 94 crops in 71 countries. Across the world, 341 unique cases of herbicide resistance among 77 weed species have been reported in wheat crops, out of which 68, 19, 10, and 9 weeds have evolved herbicide resistance against acetolactate synthase (ALS) inhibitors, acetyl CoA carboxylase (ACCase) inhibitors, synthetic auxins, and photosystem II (PS II) inhibitors (ureas and amide), respectively [36].

Earlier, herbicide resistance in India was restricted to only *P. minor* in the north-western regions against isoproturon, a substituted urea photosystem II inhibiting herbicide. This resulted in a severe reduction in the wheat yield over 0.8–1.0 million hectares wheat cultivated [13,14,37–39]. At that time, rigorous efforts were made to quantify possible reasons for this evolution of herbicide resistance and reports have shown that over-reliance on this herbicide for more than 10–15 years along with faulty spray techniques, under dosing, poor seed replacement, and the adoption of monotonous RWCS [13,17,20,37,40], have resulted in the widespread occurrence of resistance in *P. minor*. The problem of herbicide resistance has been further aggravated by the evolution of multiple resistance against newly recommended herbicides to control isoproturon-resistant *P. minor* [40–42], namely clodinafop, pinoxaden (ACCase inhibitors), sulfosulfuron (ALS inhibitor). Biotypes resistant to clodinafop exhibited cross-resistance to fenoxaprop (fop-group), tralkoxydim (dim-group), and pinoxaden (den-group), and sulfosulfuron-resistant biotypes demonstrated cross-resistance to mesosulfuron and pyroxsulam [11,17].

Further, a new problem arose in the north-western IGPs of the Haryana and Punjab regions during recent years with the evolution of metsulfuron resistance in *R. dentatus*, mainly in the RWCS. It is the dominant broadleaf weed of rabi season, and besides wheat, it infests rabi cereals such as barley [43], oat [44], oilseed crops including rapeseed mustard [45], and fodder crops including Egyptian clover [46]. In the eastern zone of Haryana, where the RWCS was dominant [38], *R. dentatus* infested about 45% of wheat fields, as compared to 15% in the western zone (non-RWCS). Furthermore, to tackle the isoproturon-resistant *P.*

*minor* biotypes, dependence on grassy herbicides such as clodinafop, pinoxaden, and sulfosulfuron was sharply increased, which also favored the dominance of BLWs, especially *R. dentatus*. The development of herbicide resistance in *R. dentatus* to metsulfuron-methyl has been reported and confirmed recently under pot and field conditions [47–51]. The first case of herbicide resistance in *R. dentatus* was reported against Group B/2 herbicides, known as ALS inhibitors (inhibit acetolactate synthase enzyme), i.e., metsulfuron-methyl [52].

Studies have demonstrated cross-resistance to iodosulfuron, triasulfuron, florasulam, iodosulfuron-methyl-sodium, mesosulfuron-methyl, halauxifen + florasulam, and pyroxsulam [11,53]. ALS is the first enzyme that completes the biosynthesis of three branched-chain amino acids (leucine, isoleucine and valine). Inhibition of ALS enzyme leads to the starvation of plant from lack of these three amino acids and causes plant death. ALS inhibitors which inhibit the ALS enzyme are broad-spectrum, post-emergence herbicides, first commercialized in 1982 with the introduction of chlorsulfuron for broadleaf weed control in cereals such as wheat, rice, and soybeans. Due to the high selection pressure associated with high inherent biological potency even at lower doses and greater reliance, this group has surpassed all known different modes of action herbicides in terms of the number of resistant weed species [54].

The herbicide resistance mechanism could be due to the target site (TS) and non-target site (NTS). The majority of cases of resistance to ALS herbicides involve the presence of modified ALS enzymes with a reduced possibility of herbicide binding [35,54]. TS-based resistance consists of a change in the molecular target of the herbicide action, which decreases its binding to the herbicide and results in reduced herbicidal action. TS-based resistance is generally associated with the amplification/over-expression of the target gene or an alteration in the target protein [35]. In NTS, a lower amount of active herbicide reaches the target site, which is associated with reduced uptake, translocation, and enhanced herbicide sequestration or metabolism [35,54].

In India, NTS was the second case of herbicide resistance and first among broadleaf weeds. Farmers were using metsulfuron-methyl, a sulfonylurea herbicide, to control broadleaf weeds in wheat since 1998. This herbicide provided effective control of major broad-leaf weeds at a very low application dose (4 g ha$^{-1}$). The herbicide metsulfuron-methyl was very effective against *R. dentatus* for the last 15 years in the wheat crop in India, but due to sole reliance on it for controlling broadleaf weeds, *R. dentatus* became resistant to this herbicide [48]. A pot-based study also demonstrated that the biotype of *R. dentatus* from Panipat (Haryana) was not effectively controlled by metsulfuron even up to 4X dose (16 g ha$^{-1}$) [51].

Seeds of *Rumex dentatus* were collected from the major rice-wheat growing districts (Figure 2B,C) of Haryana and Punjab [47]. In Haryana, 38 biotypes of *R. dentatus* were resistant to metsulfuron at 5 g ha$^{-1}$ and mostly confined to Kaithal district (6 biotypes). In Punjab, only one biotype of *R. dentatus* collected from the Barnala district demonstrated resistance to metsulfuron [47].

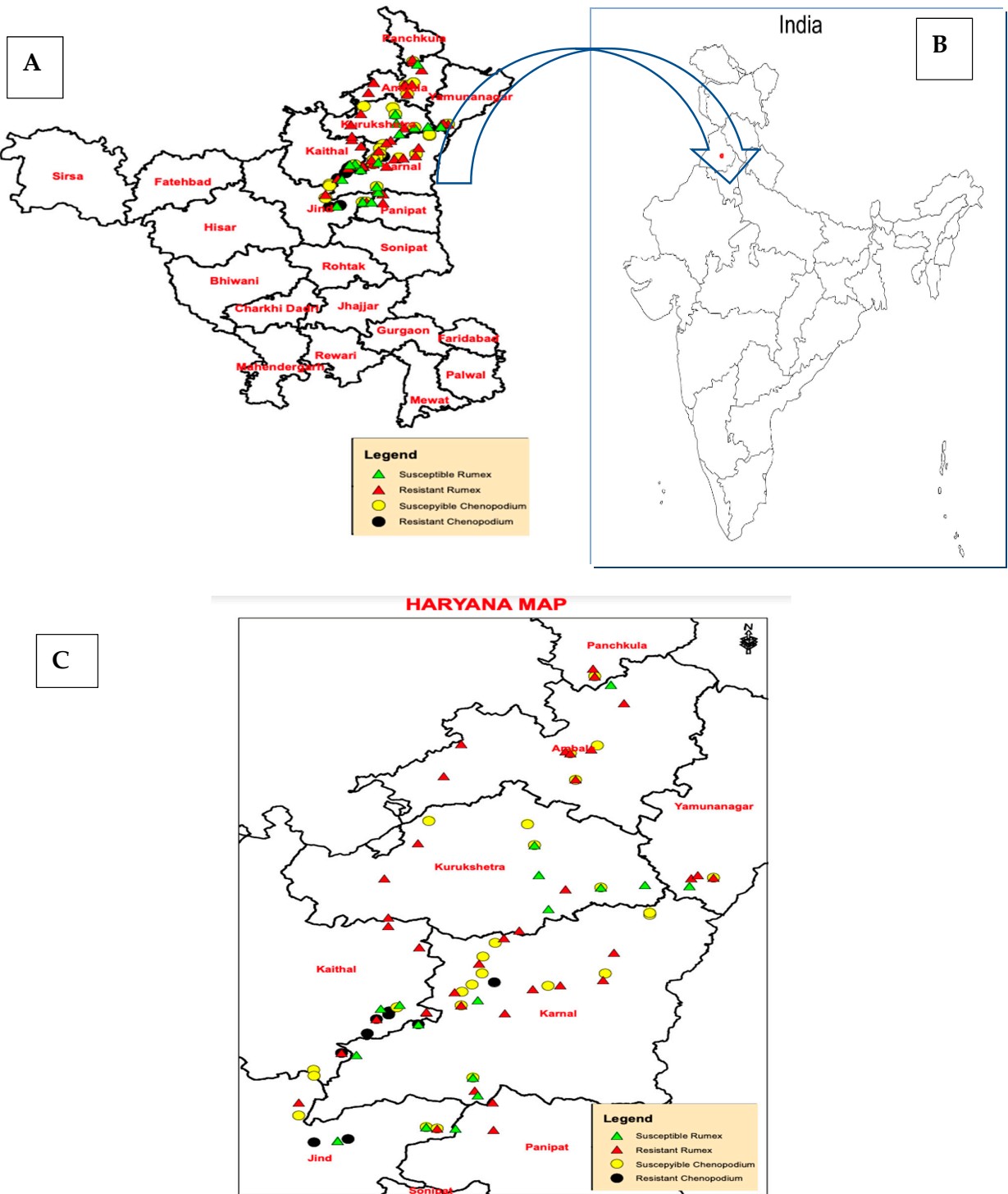

**Figure 2.** Map showing the distribution of herbicide resistance in *Rumex dentatus* and *Chenopodium album* weed species in different districts of Haryana (**A**,**C**) in India (**B**) [47].

The reasons for the failure of chemical weed control with the emergence of herbicide resistance might be sole reliance on metsulfuron, poor spray technology, or a sub-lethal dose of metsulfuron. Furthermore, a study based on a seed collection of *R. dentatus* from the major irrigated rice-wheat growing districts of Haryana and Punjab states of India [49] demonstrated that 61 *R. dentatus* biotypes from Haryana and 15 biotypes from Punjab were resistant to metsulfuron-methyl (Figure 3).

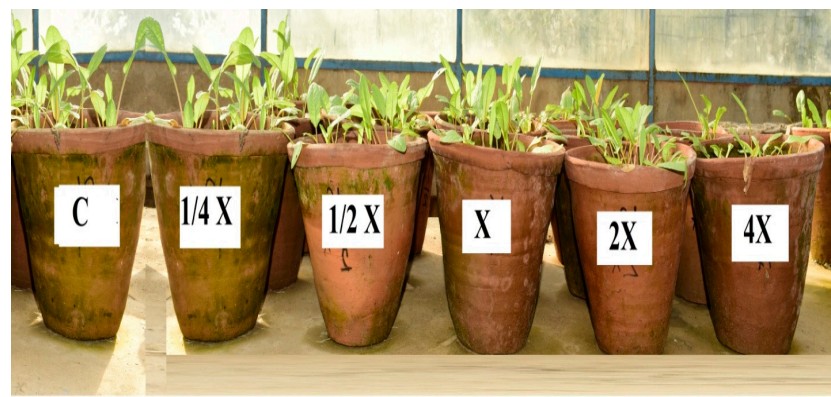

**Figure 3.** Response of *R. dentatus* biotype with graded doses (control, $\frac{1}{4}$ X, $\frac{1}{2}$ X, recommended agricultural dose (X = 5 g a.i. ha$^{-1}$), 2 X and 4 X of metsulfuron methyl (from left to right).

The evolution of herbicide resistance in multiple weeds associated with wheat is an emerging threat to the sustainability of crop production in IGPs. Farmers are facing the nuisance problem of multiple herbicide-resistant weed biotypes and, consequently, yield reduction due to the unavailability of effective alternative herbicides and lack of knowledge in tackling the issue of herbicide resistance in wheat-associated weeds (Table 1) [49,52].

**Table 1.** Status of herbicide resistance in weeds in India [11].

| Weeds | Resistance | Susceptible |
|---|---|---|
| *Phalaris minor* (Littleseed canarygrass) | Phenyl urea (Isoproturon), Sulfonylurea (sulfosulfuron, mesosulfuron), Aryloxyphenoxypropionic (Clodinafop), Cyclohexene oxime (Tralkoxydim), Phenylpyrazole (pinoxaden)and Triazolopyrimidine sulfonamide (pyroxsulam) | Flumioxazin, Pendimethalin, Metribuzin, Terbutryn, Flufenacet, and pyroxasulfone |
| *Polypogon monspeliensis* (Rabbitfoot grass) | Sulfonylurea (sulfosulfuron, mesosul-furon),Triazolopyrimidine sulfonamide (pyroxsulam) | Pendimethalin, Metribuzin Clodinafop, Fenoxaprop, Pinoxaden, Flufenacet and Pyroxasulfone |
| *Rumex dentatus* (Toothed dock) | Triazolopyrimidine sulfonamide (pyroxsulam, florasulam), Sulfonylurea (metsulfuron, triasulfuron, iodosulfuron) | 2,4-D, Carfentrazone, Pendimethalin, Flumioxazin Metribuzin & Terbutryn |
| *Chenopodium album* (Common lambsquarters) | Sulfonylurea (sulfosulfuron, metsulfuron) | 2,4-D, Carfentrazone, Flumioxazin |
| *Avena ludoviciana* (Wild oat) | Sulfonylurea (sulfosulfuron, mesosulfuron), Aryloxyphenoxypropionic (Clodinafop) | Pyroxasulfone, Flufenacet |

A diagnostic survey concluded that farmers in Haryana used lower doses of herbicides than the recommended dose, lower spray volumes, spraying at an advanced stage of weeds, and keen interest towards a single flood jet/cut or hollow cone nozzle instead of using flat fan nozzles for the application of herbicides in wheat [55]. These were possible reasons for the aggravation of multiple resistance in *P. minor* and could also have been the factors for defying the efficacy of metsulfuron-methyl against *R. dentatus*.

The world over, the first case of herbicide resistance in *R. dentatus* and the second in *Rumex* genus was reported in early 2011, when *Rumex acetosella* was reported as resistant to PS II inhibitors (hexazinone) in Canada. Recently, *Rumex obtusifolius* was reported as resistant to ALS inhibitors (B/2), and herbicides (florasulam, metsulfuron-methyl, and thifensulfuron-methyl) were reported from France in 2017 as a third case [36]. Thus, about 171 weed species have shown resistance to ALS inhibitor herbicides with 36 weed species. The mutation occurred in Trp 574 to Leu, followed by 26 weed species with amino acid substitution Pro 197 to Ser.

## 4. Agronomic Practices Influencing *Rumex dentatus* Infestation in Wheat

### 4.1. Tillage

Tillage practices play an important role in the buildup and/or shifting of weed flora and, consequently, determine crop infestation [56–58]. The sowing of wheat under a zero tillage (ZT) system reduced *P. minor* emergence [59] and biomass accumulation considerably, but favors broadleaf weed flora more than conventional tillage [21,60–62]. It has been observed that zero tillage favors the buildup of *R. dentatus* more than conventional tillage in the RWCS, which could be due to lighter seeds lower seed density of *R. dentatus* (16.71 kg hectoliter$^{-1}$) as compared to *P. minor* (61.31 kg hectoliter$^{-1}$), which are concentrated more on the soil surface after puddling in rice. Subsequently, there is greater density in wheat under puddled transplanted rice after the ZT wheat scenario [17,21]. A higher population of broadleaf weeds like *R. dentatus, Medicago denticulate*, and *Convolvulus arvensis* were found in ZT than in conventional tillage [63]. This has particular implication under the RWCS in IGP due to the development of conservation agriculture seeding machinery such as the "happy seeder" which can perform wheat seeding under heavy residue load under ZT conditions. So, the area under no-till is likely to be increased in the near future, consequently, the proliferation of *R. dentatus* will also increase.

### 4.2. Crop Rotation

Continuous cultivation of a single crop under similar cultural practices creates specific selection pressure that allows specific weeds species to become dominant due to resource availability and acquisition, and their niche [64]. Higher infestations (no./m$^2$) of the broadleaved dock (Figure 4) were found in rice-wheat (112), *fb* fallow-wheat (95), and *fb* moongbean-wheat (74), while the lowest were found in in sorghum-wheat (34) indicating more significant proliferation of dock in irrigated/saturated conditions [65]. The reduction in its population (Figure 4) under sorghum-wheat could be due to allelopathic effect of sorghum and varying patterns of resource competition [65,66].

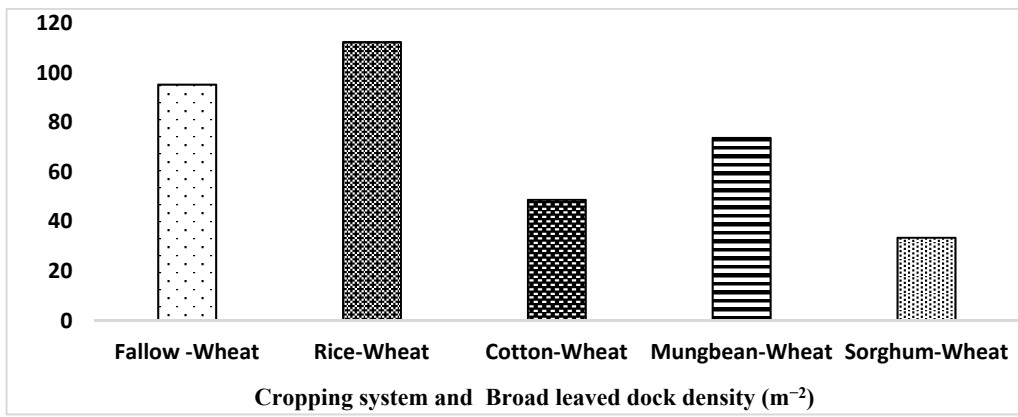

**Figure 4.** Effects of wheat-based cropping systems on broad-leaved dock density (m$^{-2}$) in wheat under weedy check conditions [65].

Regardless of herbicides, cropping system diversity is recognized as a proactive weed resistance management strategy and an important measure for diversifying weed communities and rotating selection pressure [66].

### 4.3. Paddy Straw Management (Burning/Retention)

The majority of the farmers in the north-western IGP burn the residues/straw of rice for the proper seed drilling/sowing of the succeeding wheat crop [8,9]. The burning of straw leads to a remarkable fluctuation in soil temperature that influences germination and the emergence of weeds on the soil surface [8]. However, paddy straw at surface mulch at a rate of 4 and 6 t ha$^{-1}$ reduced the emergence of *R. dentatus* by 78 and 88%, respectively, compared to controls (without mulch). The reduction in emergence was increased with an increase in the level of rice residue from 8 t ha$^{-1}$ (95%) to 10 t ha$^{-1}$ (99%) at 45 days after sowing [67]. While 2.5 t ha$^{-1}$ rice residue as mulch was not effective in suppressing weeds, 5.0 and 7.5 t ha$^{-1}$ of residue reduced weed biomass by 17–55% of *R. dentatus* compared to ZT without residue [60]. Studies relating to the *R. dentatus* response to in-situ straw burning are lacking, but the burning of rice straw is known to reduce the efficacy of soil-active herbicides like isoproturon and pendimethalin [60], along with enabling greater emergence of *P. minor*. Use of rice straw as mulch in between wheat rows when sown using CA machinery like the happy seeder can reduce the emergence of *R. dentatus* by acting as a physical barrier, so instead of burning, surface retention of rice residue could play an important role in its management.

### 4.4. Nutrient Management

Fertilizer management plays a significant role in determining crop weed interference and subsequent crop yield losses. Different aspects of nutrient management in terms of application methods (broadcast or band placement), amount, time of application, and nature of fertilizer reflect the dynamics and spatial distribution of weeds [18,68]. *R. dentatus* demonstrates higher photosynthetic nitrogen use efficiency and photosynthetic energy use efficiency, to the extent of 63–72% and 17–77%, respectively, than *P. minor* at two different levels of nitrogen (0 and 120 kg ha$^{-1}$). Even at 120 kg N ha$^{-1}$ an average increase of 12.7% in photosynthetic rate was observed [69]. *R. dentatus* is a physiologically better competitor which exhibits higher resource use efficiency than *P. minor*. The higher population of *Rumex* species was associated with excessive application of organic or synthetic nitrogen fertilizers [70], which thus act as an indicator of high nitrogen in the soil. Plants of *R. obtusifolius* were reported to take nitrates from the soil more effectively than other plants, and therefore are responsible for the temporary storage of endogenous nitrate [71]. Furthermore, with an increase in fertilizer dose from the recommended dose (150:60:40 kgha$^{-1}$ as N:P:K) to 150% of the recommended rate, the efficacy of alternate herbicides (2, 4-D ester) applied against metsulfuron-methyl resistant biotypes of *Rumex dentatus* was found to decrease (Figure 5: Chaudhary unpublished data). Farmers in the northwestern Indian plains generally apply much higher N rates than the recommended rate and these conditions will favour *Rumex dentatus* growth. Hence, there is a need to optimize fertilizer rates to reduce the efficacy of alternative herbicides against *R. dentatus* resurgence.

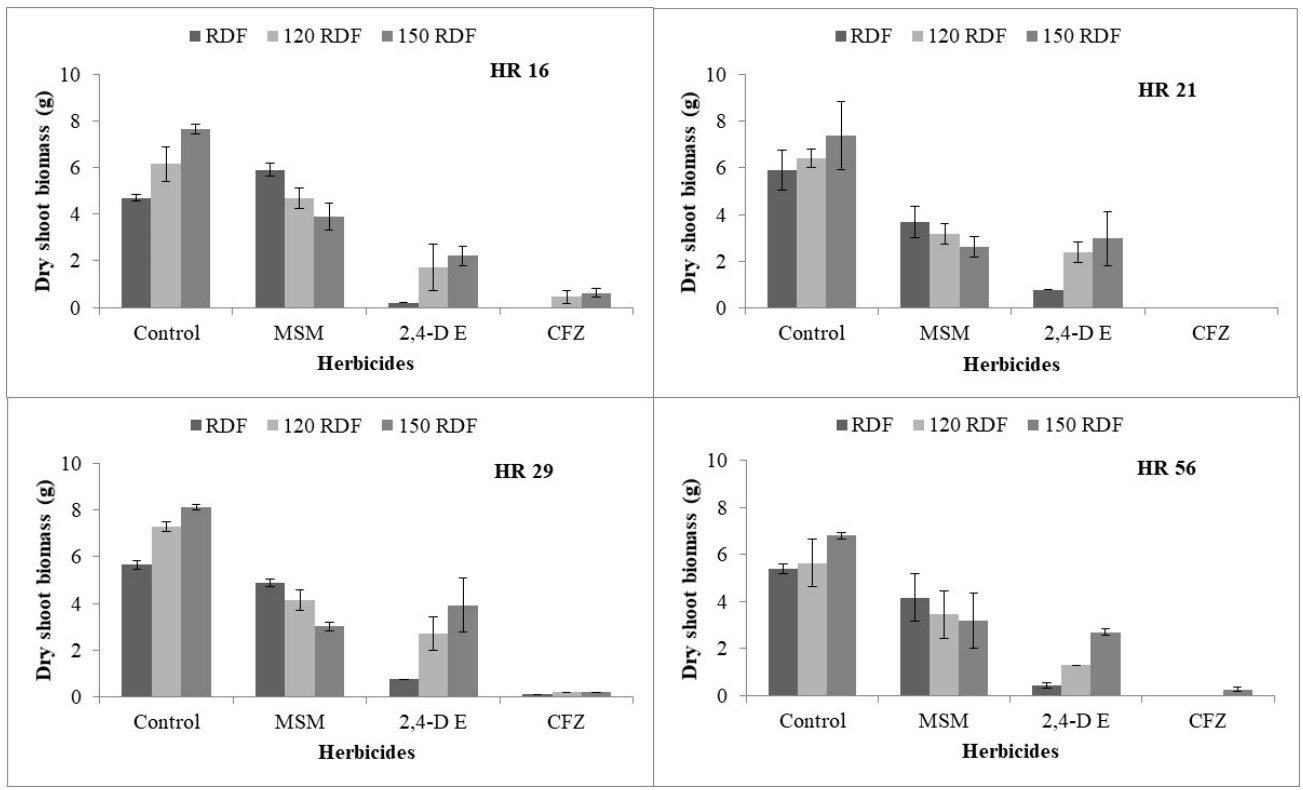

**Figure 5.** Effects of the recommended dose of fertilizer (RDF), 120%, and 150% of RDF on the efficacy of metsulfuron-methyl (MSM), 2,4 D ester (2,4 D-E), and carfentrazone (CFZ) on the growth of four resistant biotypes of *Rumex dentatus*.

### 4.5. Flooding/Irrigation Management

The emergence and growth of *R. dentatus* was not inhibited by flooding duration and a significant reduction was recorded only with 40 days of flooding [23]. Even a flooding duration of 80 days resulted in reduced *R. dentatus* emergence by 65 to 35%, compared to no flooding. *R. dentatus* was found resilient as can survive even in flooded conditions also. By contrast, *R. spinosus* is sensitive to flooding, and as a result it is not observed in the rice-wheat system. Ref. [23] also reported that the ability of *R. dentatus* to tolerate flooding conditions is the reason for its proliferation in the RWCS (high moisture conditions) as compared to other rotations. The better survival of *R. dentatus* seeds under flooded paddy fields and the accumulation of its seeds on the surface during puddling due to lower seed density facilitating floating thereby causes it to emerge in greater numbers when the succeeding wheat is sown under ZT conditions. Therefore, flooding may not help in lowering *R. dentatus* emergence. Still, deep tillage has a tendency to bury seeds to a greater depth (>4 cm), where seeds fail to emerge, resulting in lower infestation in the next season's wheat crop [23]. Frequent tillage should be avoided, as it may further enrich the surface seed bank by bringing back the seeds from the deeper soil layer. *Rumex* seeds have been reported as sensitive to moisture stress as germination is inhibited at osmotic stress higher than 0.5 mPa; however, the seed can germinate under high salinity conditions [33].

### 5. Management of Metsulfuron-Methyl-Resistant *Rumex dentatus*

Dependence on a single agronomic practice is not effective for the long-term management of any weed species. Therefore, the integrated weed management (IWM) approach is better for the management of *R. dentatus*. The IWM approach should include cultural, mechanical, chemical, and biological practices.

Cultural practices include competitive cultivars, sowing, seed rate, row spacing, the stale bed method, band application of fertilizers, uprooting weeds before seed set, crop diversification, and retention of previous crop residue on the soil as mulch to prevent the

emergence of *R. dentatus* [17,65]. ZT with crop residues could enhance weed seed predation and seed decay because (1) a more significant proportion of weed seeds remains on the soil surface where they are more prone to seed predation (2) residues provide a desirable habitat for seed predators and decay agents (3) improved soil characteristics under ZT could facilitate more density of seed predators and decay agents [72]. However, the continuous flooding and tillage may induce the mortality of potential seed predators [72].

Biological control of weeds is an eco-friendly approach that helps to minimize the weed population. Many micro-organisms or bio-agents can be potentially used for the effective management of *R. dentatus.* For example, the fungi *Alternaria alternata* can cause 70% mortality in *R. dentatus* and pathogenicity, increasing the spore concentration and dew period [73]. A spore concentration of $10^7$ conidia $mL^{-1}$ in 20% canola oil emulsion under 100% complete saturated atmospheric conditions resulted in 100% mortality of *R. dentatus* when applied at the 3–4 leaf stage. The mortality rate was lower when applied at 10–12 leaf stage. Maximum mortality of *R. dentatus* was found at 25–30 °C. The other potential fungal strain against *Rumex dentatus* is *Nigrospora oryzae* YMM4, which can be used to produce mycoherbicides [74].

The metabolites of *Drechslera* spp. (*D holmii, D biseptata*, and *D australiensis* in the original and 50% diluted forms reduced *R. dentatus* seedling germination, roots, and shoot biomass by 5–56, 68–88, and 15–83%, respectively [75]. Furthermore, a study from Pakistan demonstrated that the aqueous extract of sunflower leaves reduced the *R. dentatus* count by 46 and 67% with 80 and 100% leaf extract concentrations, respectively [76]. So these approaches need integration for effective control of this weed under location-specific conditions.

The chemical approach consists of the application of pre-and post-applied herbicides alone or in mixtures. The development of resistance against metsulfuron-methyl narrows down the options for broadleaf weed control in wheat. As for the control of these weeds in wheat, only three major herbicides are available and used in India (2,4-D ester/amine/sodium salt, metsulfuron-methyl, and carfentrazone-ethyl) [77]. The efficacy of these herbicides (2,4-D and carfentrazone) is very much stage-dependent, especially the contact herbicide carfentrazone-ethyl that is less effective at advanced stages of weeds and is unable to control subsequent emerging weeds due to its poor residual effect. Carfentrazone and 2,4-D provided effective control of the resistant *R. dentatus* biotype about 87 and 98%, respectively, as compared to 17% control with metsulfuron-methyl at the recommended herbicide rate [51]. Compared to metsulfuron, 2,4-D demonstrated poor efficacy for the control of sensitive *R. dentatus* and the formulation-based order of effectiveness was 2,4-D amine salt followed by 2,4-D ester salt, and 2,4-D sodium salt [52].

Herbicides like carfentrazone, metsulfuron-methyl plus carfentrazone, and halauxifen methyl plus florasulam were found effective for the effective control of *R. dentatus.* However, the halauxifen methyl plus florasulam mixture was not effective for the control of resistant *Rumex* (Chhokar unpublished data). Pendimethalin and metribuzin provided effective control of *R. dentatus* of about 98–100% and 68–92%, respectively. Pre-emergence tank-mix application of pendimethalin 0.75–1.0 kg $ha^{-1}$ plus metribuzin 0.175 kg $ha^{-1}$ or sequential application of pendimethalin 0.75–1.0 kg $ha^{-1}$ at pre-emergence and sulfosulfuron 0.018 kg $ha^{-1}$ at post-emergence could be adopted for broad-spectrum weed control in wheat [78]. Metsulfuron plus carfentrazone and 2,4-D amine/ester provided effective control of *R. dentatus* as compared to mesosulfuron plus iodosulfuron, sulfosulfuron plus metsulfuron, and metsulfuron [50]. These herbicides can be served as alternate herbicidal options and should be used and rotated for the management of resistant biotypes of *R. dentatus* and to prevent further proliferation of herbicide resistance (Table 2).

**Table 2.** Alternate herbicides available to control ALS-based herbicide resistant *R. dentatus* [52,77,79].

| Herbicide Name | Dose (g ha$^{-1}$) | Mode of Action |
|---|---|---|
| 2, 4-D sodium salt | 500 | Synthetic auxin |
| 2, 4-D ethyl ester | 500 | Synthetic auxin |
| 2,4-D amine | 750 | Synthetic auxin |
| Carfentrazone ethyl | 20 | Protoporphyrinogen oxidase inhibitor |
| Metsulfuron methyl + Carfentrazone ethyl (premix) | 25 | ALS inhibitor + Protoporphyrinogen oxidase inhibitor |
| Bromoxynil + MCPA | 490 | Photosystem II inhibitor + synthetic auxin |
| Isoproturon | 1000–1250 | Photosystem II inhibitor |
| Metribuzin | 300 | Photosystem II inhibitor |
| Terbutryn | 1000 | Photosystem II inhibitor |
| Dicamba | 240–360 | Synthetic auxin |

Weed populations can be reduced by the sowing of wheat with higher seed rate (125 kgh$^{-1}$) under zero tillage + residue retention (8 t ha$^{-1}$) coupled with a pre-emergence herbicide mixture (pendimethalin + metribuzin at 1.5 + 0.210 kg ha$^{-1}$) beneath the mulch [80].

Knowledge of the mechanisms of herbicide resistance in weeds is instrumental in designing new strategies for weed management and to discourage the further evolution of herbicide resistance. In IGP, resistance development in *P. minor* against Group C (medium risk) isoproturon herbicide took 10–15 years, but after that in the case of group A/group B (high risk), cross-resistance was developed within a very short time [81]. So, these points should be kept in mind while introducing new herbicide chemistry for *R. dentatus* as well. There should be a rotation or mixture of high- and low-risk groups to delay herbicide resistance development in weed species.

## 6. Climate Change and Weed Dynamics

Since 1958, Global $CO_2$ has increased by 24% to a current level of 390 ppm, and will reach 550 ppm by 2050, with global surface temperature to rise from 1–4 °C [82]. Ongoing increase in atmospheric $CO_2$ is likely to affect the photosynthetic rate in C$_3$ plants (both *R. dentatus* and its associated wheat crop) by increasing the $CO_2$ concentration gradient from the air to the leaf interior and decreasing the loss of $CO_2$ via photorespiration [83,84]. Since elevated $CO_2$ favors the C$_3$ pathway in both crops and weeds, the competitiveness of crops against weeds would be bounded by resource availability such as nutrients, water, light, etc. [84]. Most studies advocate that crop–weed interactions (C$_3$-C$_3$) differ significantly under $CO_2$ enriched environments and favors C$_3$ weed flora more than crops under limited nutrient or water availability conditions [85,86]. Climate change is bound to influence the ecology of weeds and possible implications for their management. Weeds, by virtue of their greater genetic diversity, have better adaptability to the changing climate than do crops. Weed management is likely to become more complex in the future due to increased invasiveness, weed shifts, and greater chances of herbicide resistance developments under a changing climate, and are likely to influence population dynamics of weed species, crop–weed interactions, and their gravity of infestation [84–88]. Furthermore, herbicide efficacy, along with higher chances of defying herbicidal action, is a matter of concern [87,88]. Studies of under Indian conditions that correlate the quantification of competitive advantages associated with elevated $CO_2$ and temperature for *Rumex dentatus* under resource-limiting environments are lacking.

### 7. Future Prospects

Sole dependence on herbicides along with a mono-cropping sequence led to the shift towards harder-to-control weeds and the rapid development of herbicide resistance, which could threaten wheat productivity. Therefore, it is essential to determine the current status of herbicide resistance/poor efficacy of different herbicides against *R. dentatus* in farmers' fields through a systematic survey. Also, there is need to understand the resistance profile of different biotypes of *R. dentatus* from the rice-wheat belt of IGPs through bioassay studies. In addition, there is a need to identify effective alternative herbicidal options for weed management so that the problem of herbicide resistance may be tackled effectively.

The mechanism of resistance in *R. dentatus* to metsulfuron-methyl is either unknown or not entered in the database, and also there is no record of the differences in the fitness or competitiveness of these group B/2 resistant biotypes compared to susceptible biotypes. Therefore, there is a need to research these untapped areas to manage the resistance problem in *R. dentatus* in a timely and efficient manner. Further, there is a need to understand the biological fitness of resistant biotypes in relation to future scenarios (such as elevated $CO_2$ or temperature under limited resource availability, i.e., nutrient and moisture) in comparison to susceptible one. The development of new chemicals as alternative herbicides supplemented with non-chemical methods of weed management would be very effective against metsulfuron-resistant *R. dentatus*. Continuous use of a single herbicide could be discouraged and a rotational use of herbicides with proper spray techniques could be advocated in order to avoid and delay the chances of the development of resistance/cross-resistance in *R. dentatus* in India. Therefore, future studies need to evaluate the compatibility/suitability between different broad-leaf herbicides and broad-leaf and grass herbicides. Further, the similar maturity time of *R. dentatus* and wheat provides an opportunity for a weed-seed destructor-based assembly that can be attached to combine harvester to limit the replenishment and enrichment of the weed seed bank.

### 8. Conclusions

Earlier in India, the herbicide resistance problem was confined to only *Phalaris minor*, but with the intensive use of solo metsulfuron-methyl to control broad leaf weeds, it has now resulted in the aggravation of the resistance problem with the addition of one more case of *R. dentatus*. Under Indian conditions, there are only few broadleaf weed herbicides for wheat in the market, and not all of them are compatible in a mixture with grass herbicides. Thus, there is a need to explore more herbicidal chemistries with high compatibility with grassy herbicides so that they can be used to increase the spectrum of weed control under diverse agro-climatic conditions. Long-term effective resistance management plans should comprise of integrating chemical and non-chemical means of weed management with weed biology knowledge. Integrated weed management strategies must be developed to prevent the spread of herbicide-resistant weeds for sustainable wheat production. Furthermore, intensive and periodical scouting is required to broaden the knowledge of the geological distribution, resistance mechanisms, biological fitness, and dynamics of the growth response in relation to the fickle climate to rectify the gravity of resistance with respect to the development of the strategic management of herbicide-resistant weeds.

**Author Contributions:** Conceptualization, A.C., R.S.C., writing—original draft, A.C., S.D., S.K. (Simerjeet Kaur); writing—review and editing, A.C., R.S.C., P.K., T.M.P., S.S.P.; data curation, A.C., R.S.K., S.K. (Surender Kumar), P.K. checked and corrected the final draft. All authors have read and agreed to the published version of the manuscript.

**Funding:** This research received no external funding.

**Institutional Review Board Statement:** Not applicable.

**Acknowledgments:** The authors are highly thankful to the Agronomy Department of Chaudhary Charan Singh Haryana Agricultural University (Hisar), Punjab Agricultural University (Ludhiana) and the ICAR-Indian Institute of Wheat and Barley Research (Karnal).

**Conflicts of Interest:** The authors declare no conflict of interest.

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
