# Peer review of "Herbicide Resistance to Metsulfuron-Methyl in Rumex dentatus L. in North-West India and Its Management Perspectives for Sustainable Wheat Production"

_sustainability, doi:10.3390/su13126947_

Round 1

Reviewer 1 Report

Article is very interesting, many courses of literature were used. I find a little bit distraction: f. e. In section 2.( Distribution and importance of Rumex dentatus) many information about weeds in general and lack about R. dentatus. Maybe it is worth integrate in one section Importance  biology and ecology of this weed. Reporting overall should be a little bit more consistent. 

Author Response

1) Article is very interesting; many courses of literature were used.

  Reply:   Thank you for your kind words.

2) I find a little bit distraction: f. e. In section 2. (Distribution and importance of Rumex dentatus) many information about weeds in general and lack about R. dentatus. Maybe it is worth integrate in one section Importance biology and ecology of this weed.

Reply: We have integrated ‘Economic importance, biology and ecology of R. dentatus’ as suggested by you.

3) Reporting overall should be a little bit more consistent.

Reply: We have modified the manuscript to make it more coherent.

Reviewer 2 Report

Peer review on manuscript

“Herbicide resistance to metsulfuron-methyl in Rumex dentatus in north-west India and its management perspectives for sustainable wheat production: A review”

This review article aggregate the current state of various aspects very important issues related to herbicide resistance in weeds that strongly affects the rice and wheat cropping in India. In particular, authors analysed the a herbicide resistance in Rumex dentatus, essential agronomic factors that affect its population and propagation across cropping systems, as well as strategies to manage the infestation of resistant population. The analysis of the issues considered in the article is carried out in sufficient detail, and the conclusions formulated by the authors are fully justified. The article is well-organized and well-written. I have no objection to publishing this article.

To improve the paper, I recommend authors to make some additions in the paper. First, I believe that it is advisable for the authors (in a separate paragraph) to express their opinion on how the results of their article can be generalized to other geographic regions. Second, it would be very helpful if the authors provided their views on how climate change will affect weed spread?. What adaptation measures could be proposed in this regard?

Author Response

1) This review article aggregate the current state of various aspects very important issues related to herbicide resistance in weeds that strongly affects the rice and wheat cropping in India. In particular, authors analysed the a herbicide resistance in Rumex dentatus, essential agronomic factors that affect its population and propagation across cropping systems, as well as strategies to manage the infestation of resistant population. The analysis of the issues considered in the article is carried out in sufficient detail, and the conclusions formulated by the authors are fully justified. The article is well-organized and well-written. I have no objection to publishing this article.

Reply: Thank you for your kind words and we are thankful to you for thorough reading of the review.

2) To improve the paper, I recommend authors to make some additions in the paper. First, I believe that it is advisable for the authors (in a separate paragraph) to express their opinion on how the results of their article can be generalized to other geographic regions. Second, it would be very helpful if the authors provided their views on how climate change will affect weed spread?. What adaptation measures could be proposed in this regard?

Reply: 

Your suggestions are relevant and we have incorporated these two aspects.

Opinion on how the results of their article can be generalized to other geographic regions:- 

Rumex dentatus is one of the major weeds infesting winter crops especially wheat crop in North India. Our review suggests that R. dentatus has evolved resistance against several ALS inhibitors and evolution of resistance in R. dentatus is reported at farmers’ fields in two major wheat-growing states (Punjab and Haryana) only. This resistance is mainly associated with monoculture of rice-wheat system and non-adoption of herbicide rotation. A strong selection pressure was imposed on the R. dentatus weed which might have resulted in evolution of herbicide resistance in it. It is emerging as a severe problem under conservation tillage practices and its build up is more in zero-till wheat as compared to conventional till wheat crop. The herbicide resistance in R. dentatus after P. minor in these northern India is an alarming signal for sustainable wheat production. The spread of resistant weeds from these states to other geographical regions where wheat crop is grown in winter season needs to be restricted by following quarantine measures. Preventive measures such as encouraging the farmers to use certified seeds, restricted movement of crop seeds and farm yard manure contaminated with seeds of resistant biotypes to the new area should be followed. Manual weeding can be adopted to rogue out the plants before seed setting which survived even after application of herbicide.

3) How climate change will affect weed spread and adaptation measure

Reply: Separate paragraph is added and discussed in detail

Reviewer 3 Report

Paper is related to herbicide resistance issues in Rumex dentatus. Authors wrote a review with different perspectives and at certain point provided an updated about the resistance issue in India. There are some changes I suggest to improve the manuscript.

In the abstract section please provide a conclusion or future perspective, the way it is does not give a real perspective about the herbicide resistance issue and only a general statement is given (L29).

In the manuscript I would suggest to give a chapter of Herbicide resistance mechanisms involved not only in R dentatus (if any paper exist) but other ALS resistant accessions relevant to wheat cropping systems. Despite some resistance mechanism are described in the ms, a special section is needed since as the title mentions paper is about herbicide resistance.

- Authors need to give a deep review of the manuscript in regards to terminology. Please and accordingly, use at first appearance the first full scientific name with the authority and then stick to this terminology. See for instace R. dentatus in L20 versus L23. This includes all scientific names through the manuscript.

-Please review through the ms the nomeclature and correct accordingly. See t/ha versus t ha-1

-L64 correct expression single mode of herbicides.

-L68. This paragraph I would rather name it as Economical importance of Rumex dentatus or Economical losses…since it talks more about numbers than distribution of Rumex dentatus. Please correct accordingly.

Figure 1. Please redo the graph adding a letter A and B respectively and rewriting the caption. In the same sense, can authors provide a better photograph? Since these do not have a good resolution. In Rumex dentatus seeds please provide a scale.

Figure 2. Please consider sticking together 1 and 2 and make A,B,C,D since it would stand alone in a better way. Please provide a better picture if available and crop the second image to have the horizon line straight.

Please correct in L147-149 the meaning of herbicide resistance weeds, since the weeds are not resistant to herbicides but the accessions (in this sense). E.g., herbicide resistance weed accessions.

L156. Please correct sentence with PSII inhibitor.

L157 please double check the data provided.

L200. Please replace the letter X by the times symbol through the ms. This includes Fig 4.

Figure 3. Need a better resolution figure. Consider placing fig 3 and 5 together.

Figure 4. Please provide a better resolution image or do not enlarge the one in the ms.

Figure 5 Delete caption in black and the top and rewrite caption. Change the legends symbol, since they are the same when printing b&w

Table 1. Please be consistent with the common names. Specify what Sr no. means.

Figure 7. Need a better figure.

Author Response

1) Paper is related to herbicide resistance issues in Rumex dentatus. Authors wrote a review with different perspectives and at certain point provided an updated about the resistance issue in India. There are some changes I suggest to improve the manuscript.

Reply: We are thankful to you for thorough reading of the review.

2) In the abstract section please provide a conclusion or future perspective, the way it is does not give a real perspective about the herbicide resistance issue and only a general statement is given (L29).

Reply:  Abstract is modified as suggested

3) In the manuscript I would suggest to give a chapter of Herbicide resistance mechanisms involved not only in R dentatus (if any paper exist) but other ALS resistant accessions relevant to wheat cropping systems. Despite some resistance mechanism are described in the ms, a special section is needed since as the title mentions paper is about herbicide resistance.

Reply: Required changes are performed (Line 270 to 278)

4) Authors need to give a deep review of the manuscript in regards to terminology. Please and accordingly, use at first appearance the first full scientific name with the authority and then stick to this terminology. See for instance R. dentatus in L20 versus L23. This includes all scientific names through the manuscript.

Reply: Required changes are performed

5) Please review through the ms the nomenclature and correct accordingly. See t/ha versus t ha-1

Reply:  All units are written uniformly like t ha-1 (line 25 in abstract)

6) L64 correct expression single mode of herbicides.

Reply: The said sentence is modified

7) L68. This paragraph I would rather name it as Economical importance of Rumex dentatus or Economical losses…since it talks more about numbers than distribution of Rumex dentatus. Please correct accordingly.

Reply:  The section is renamed as ‘Economical importance of Rumex dentatus’ as suggested by you.

8) Figure 1. Please redo the graph adding a letter A and B respectively and rewriting the caption. In the same sense, can authors provide a better photograph? Since these do not have a good resolution. In Rumex dentatus seeds please provide a scale.

Reply: Required changes are performed as per suggestions

9) Figure 2. Please consider sticking together 1 and 2 and make A,B,C,D since it would stand alone in a better way. Please provide a better picture if available and crop the second image to have the horizon line straight

Reply: Required changes are performed 

10) Please correct in L147-149 the meaning of herbicide resistance weeds, since the weeds are not resistant to herbicides but the accessions (in this sense). E.g., herbicide resistance weed accessions.

Reply: Required changes are performed

11) L156. Please correct sentence with PSII inhibitor.

Reply: Required changes are performed (photosystem II inhibitor)

12) L157 please double check the data provided.

Reply: Required changes are performed 

13) L200. Please replace the letter X by the times symbol through the ms. This includes Fig 4.

Reply: Required changes are performed (1/4 times, ½ times, recommended dose, 2 times and 4 times)

14) Figure 3. Need a better resolution figure. Consider placing fig 3 and 5 together.

Reply: Required changes are performed (figure 3 and 5 replaced with better resolution figure and placed together)

15) Figure 4. Please provide a better resolution image or do not enlarge the one in the ms

Reply: Required changes are performed (size is reduced)

16) Table 1. Please be consistent with the common names. Specify what Sr no. mean

Reply: Required changes are performed (sr. No. deleted from the table 1,2)

17) Figure 5 Delete caption in black and the top and rewrite caption. Change the legends symbol, since they are the same when printing b&w

Reply: We have remove this figure accordingly

Round 2

Reviewer 3 Report

Manuscript has been improved substantially, however authors did not incorporate my suggestions as they are commenting. 

Fig 1A needs to add the scale, authors stated they added one (see comment 8). However in the present ms i do not see it.

L161 change to

 , a substituted urea photosystem II inhibiting-herbicide.

Figure 2. Again, here we have the issue of resolution, I can not read what is in the map (A), or in the label of C figure. Need better images.

Figure 3. What I meant was to substitute the letter "X" that authors are using for the "×" multiplication sign. This affects to figure caption and legends of Figure 3 and the whole ms.

Table 1 add the missing common names.

L162. what authors mean by 0.8-1.0 m ha? 80 cm?

Table 2.

Modify to Dose (g ha-1) and correct accordingly.

Author Response

1) Fig 1A needs to add the scale, authors stated they added one (see comment 8). However in the present ms I do not see it.

     Required changes are performed 

2) L161 change to a substituted urea photosystem II inhibiting-herbicide.

    Required changes are performed

3) Figure 2. Again, here we have the issue of resolution, I can not read what is in the map (A), or in the label of C figure. Need better images.

  Required changes are performed with better resolution figure/map

4) Figure 3. What I meant was to substitute the letter "X" that authors are using for the "×" multiplication sign. This affects to figure caption and legends of Figure 3 and the whole ms.

    Here X denotes agricultural rate of herbicide (recommended dose) as shown on different pots, It’s not  multiplication sign.  Required changes are performed 

5) Table 1 add the missing common names.

   Common names are added in Table 1

6) L162. what authors mean by 0.8-1.0 m ha? 80 cm?

   The said sentence is modified and m ha denotes million hectares

7) Table 2. Modify to Dose (g ha-1) and correct accordingly.

Required changes are performed as per suggestions